# FOUNDATIONFORENSICS: TRACEBACK BACKDOOR ATTACKS FOR VISION FOUNDATION MODELS

## ABSTRACT

Foundation models are typically pre-trained on uncurated unlabeled data collected from various domains on the Internet. As a result, they are fundamentally vulnerable to backdoor attacks, where an attacker injects carefully crafted poisoned inputs into the pre-training data via hosting them on the Internet. A backdoored foundation model outputs an attacker-desired embedding vector for any input with an attacker-chosen trigger. In this work, we propose FoundationForensics, the first forensics method to trace back poisoned pre-training inputs for foundation models *after* a backdoor attack has happened and a trigger-embedded input has been detected. Our FoundationForensics first calculates a maliciousness score for each pre-training input by quantifying its contribution to the foundation model's backdoor behavior for the detected trigger-embedded input and then detects the pre-training inputs with outlier maliciousness scores as poisoned. We theoretically analyze the security of FoundationForensics and empirically evaluate it on single-modal and multi-modal foundation models, three datasets, four existing backdoor attacks, and seven adaptive ones. Our results show that FoundationForensics can accurately traceback the poisoned pre-training inputs for foundation models.

## 1 INTRODUCTION

Vision foundation models–such as CLIP (Radford et al., 2021), SAM (Kirillov et al., 2023), and Dino (Caron et al., 2021)–produce general-purpose *embedding vectors* for images inputs. A service provider (e.g., OpenAI, Google, and Meta) often first collects a vast amount of *unlabeled* data (called *pre-training data*)–such as images and image-text pairs–from various public Internet *domains* such as websites and social media platforms. The collected, often uncurated pre-training data is then used to pre-train a foundation model via self-supervised learning (Chen et al., 2020; Radford et al., 2021; Devlin et al., 2019). After pre-training, foundation models can be used to build various downstream applications from classification to generative AI, such as text-to-image generative models (Rombach et al., 2022; Ramesh et al., 2022) and multi-modal large language models (Liu et al., 2024a).

However, foundation models pre-trained on uncurated Internet data are fundamentally vulnerable to backdoor attacks (Carlini & Terzis, 2021; Liu et al., 2022; Zhang et al., 2024; Xu et al., 2024). In particular, an attacker can inject carefully crafted *poisoned inputs* into the pre-training data via hosting them on public Internet domains (Liu et al., 2022; Carlini et al., 2023). The backdoored foundation model outputs an *attacker-desired* embedding vector for any input with an attacker-chosen *backdoor trigger*, while the embedding vectors for inputs without the backdoor trigger are unaffected. The backdoor trigger could be, for example, a colored square in an image input. An attacker-desired embedding vector is typically the embedding vector of an attacker-chosen input (called *target input*). Such a backdoored foundation model leads to a single-point-of-failure of the AI ecosystem since all downstream applications inherit the backdoor behavior.

Defenses against backdoor attacks to foundation models can be categorized into *prevention* (Bansal et al., 2023; Liu et al., 2022; Yang et al., 2023), *detection* (Feng et al., 2023; Ma et al., 2023), and *forensics* (Liu et al., 2024b), which are complementary and can be combined in a defense-in-depth fashion. Prevention re-designs the pre-training algorithm or filters poisoned inputs to ensure a backdoor-free pre-trained foundation model but often sacrifices its utility substantially (Liu et al., 2022). Detection identifies backdoored foundation models (Feng et al., 2023) or trigger-embedded inputs (Ma et al., 2023). After detecting a backdoor attack, forensics methods are applied to analyze

the root cause and recover the foundation model from the attack. For instance, given a detected trigger-embedded input, Mudjacking (Liu et al., 2024b) can remove the backdoor from a foundation model while maintaining its utility by strategically adjusting its parameters. However, Mudjacking can not trace back the root cause (i.e., the poisoned inputs) of a detected backdoor attack. Tracing back the poisoned inputs and the Internet domains where they are collected from is crucial for forensic analysis. The identified poisoned inputs and Internet domains serve as step stones for forensics analysts to identify the attackers/criminals.

**Our work:** In this work, we propose FoundationForensics, the first forensics method to trace back the poisoned inputs in a detected backdoor attack to foundation models. The tracing back process is shown in Figure 1. Following Mudjacking (Liu et al., 2024b), we assume a *backdoor instance* $(x_b, x_r)$ has been detected, where $x_b$ is a trigger-embedded input and $x_r$ is a clean *reference input*. $x_b$ and $x_r$ have different semantics (e.g., they include different objects) but the backdoored foundation model outputs similar embedding vectors for them.

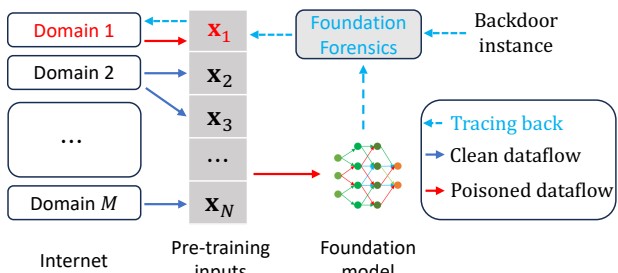

Figure 1: Given a backdoor instance, our FoundationForensics traces back the poisoned inputs and attack source for a backdoored foundation model. $\mathbf{x}_1$ is a poisoned input.

The backdoor instance $(x_b, x_r)$ can be detected manually or automatically (Ma et al., 2023; Chou et al., 2020; Gao et al., 2019).

Given a backdoor instance $(x_b, x_r)$, our FoundationForensics traces back the poisoned inputs in the pre-training data via two key steps: 1) calculating an maliciousness score for each pre-training input, and 2) detecting poisoned inputs via outlier analysis of the maliciousness scores. In the first step, FoundationForensics aims to assign an maliciousness score to each pre-training input, quantifying its contribution to the cosine similarity between the embedding vectors of $x_b$ and $x_r$ in the given backdoor instance. We propose to expand the pre-training process by tracking and aggregating the contribution of pre-training inputs to the foundation model parameters and thus the cosine similarity across pre-training epochs, thereby assigning maliciousness scores.

In the second step, our FoundationForensics detects the pre-training inputs with outlier maliciousness scores as poisoned inputs. Our intuition is that poisoned inputs would have abnormally large maliciousness scores and thus they are outliers. We use the well-known method *Median Absolute Deviation* (Pham-Gia & Hung, 2001) to detect outliers. Specifically, FoundationForensics first calculates the median $M$ of all pre-training inputs' maliciousness scores. Then, FoundationForensics calculates the absolute deviation of each pre-training input's maliciousness score from the median $M$ and determines the median (denoted as $\tilde{M}$) of the absolute deviations. Finally, FoundationForensics identifies the pre-training inputs whose maliciousness scores are larger than $M + k \cdot \tilde{M}$ as poisoned inputs, where $k$ is a hyperparameter to tune the sensitivity of the outlier detection method.

Our evaluation is two-fold. On one hand, we *theoretically* show the security of FoundationForensics against backdoor attacks. In particular, we prove that a poisoned input has a larger maliciousness score than a clean one. On the other hand, we *empirically* evaluate FoundationForensics on three vision foundation models, three benchmark datasets, four existing backdoor attacks, and eight adaptive ones. Our results show that FoundationForensics can accurately trace back the poisoned inputs under various backdoor attacks. Moreover, FoundationForensics outperforms existing forensics methods for classifiers (Shan et al., 2022; Hammoudeh & Lowd, 2022) when extended to foundation models.

In summary, our main contributions are as follows:

- We propose FoundationForensics, the first forensics method to trace back poisoned inputs in backdoor attacks to foundation models after attack detection.

- We theoretically show the security of FoundationForensics against backdoor attacks.

- We empirically evaluate FoundationForensics on multiple foundation models and datasets under various existing and adaptive backdoor attacks.

## 2 PRELIMINARIES AND RELATED WORK

**Vision foundation models:** Given an image $x$, a vision foundation model $f$ outputs a general-purpose embedding vector $f(x)$. Vision foundation models can be pre-trained on unlabeled images, known as *single-modal vision foundation models*, such as SimCLR (Chen et al., 2020) and MoCo (He et al., 2020). Alternatively, vision foundation models can be pre-trained on image-text pairs, known as *multi-modal vision foundation models*, like CLIP (Radford et al., 2021). Given a pre-trained foundation model as a general-purpose feature extractor, a developer can build various downstream applications from classifications to generative AI.

**Backdoor attacks:** A backdoored foundation model $f$ has two properties: 1) $f$ outputs an attacker-chosen embedding vector $F$ for any input $x_b$ embedded with an attacker-chosen trigger (e.g., a colored square at the bottom right corner of an image), i.e., $f(x_b) \approx F$; and 2) $f$ outputs a high-quality embedding vector for any input without the trigger, i.e., downstream applications built based on $f$ have high performance for inputs without the trigger.

An attacker can create such a backdoored foundation model via *data-poisoning* or *model-poisoning* backdoor attacks. In data-poisoning backdoor attacks (Liu et al., 2022; Carlini & Terzis, 2021; Saha et al., 2022), an attacker embeds backdoor into a foundation model via injecting poisoned inputs into its pre-training data; while in model-poisoning backdoor attacks (Jia et al., 2022; Shen et al., 2021; Zhang et al., 2023; Tao et al., 2024), an attacker embeds backdoor into a foundation model via directly editing its model parameters. Model-poisoning backdoor attacks target the supply chain of foundation models. For instance, an attacker can download a clean foundation model from Hugging Face, edit its model parameters to embed backdoor, and then republishes the backdoored foundation model on Hugging Face. When developers download the attacker's backdoored foundation model from Hugging Face and build applications based on it, the applications inherit the backdoor behavior. Therefore, model-poisoning backdoor attacks pose less threats than data-poisoning backdoor attacks. This is because developers can obtain a foundation model from a trusted service provider (e.g., Meta, OpenAI, or Google), who is unlikely to embed backdoor into its foundation model via model-poisoning backdoor attacks. In contrast, in data-poisoning backdoor attacks, attackers can publish the poisoned inputs (e.g., poisoned images or image-text pairs) on the Internet; and when a service provider collects unlabeled pre-training data from the Internet, the poisoned inputs may be collected and backdoor is embedded into a foundation model during pre-training. Therefore, in this work, we focus on data-poisoning backdoor attacks and trace back the poisoned inputs after attack detection.

Different data-poisoning backdoor attacks assume different attacker-chosen embedding vector $F$ and use different strategies to craft the poisoned inputs. For example, in PoisonedEncoder (Liu et al., 2022) that attacks single-modal vision foundation models, $F$ is the embedding vector of an attacker-chosen clean target input, while the poisoned inputs are crafted by concatenating trigger-embedded inputs with an attacker-chosen target input. Carlini & Terzis (2021) crafts image-text pairs as poisoned inputs to attack multi-modal vision foundation models, where the text can be viewed as "target label" for the corresponding image. The attacker embeds a trigger into images and modifies the corresponding texts to include the attacker-desired target label. For instance, the target label could be "a photo of dog", whose text embedding is the attacker-chosen embedding vector $F$.

**Defenses:** Defenses can be categorized into *prevention* (Bansal et al., 2023; Liu et al., 2022; Yang et al., 2023), *detection* (Feng et al., 2023; Ma et al., 2023), and *forensics* (Liu et al., 2024b). Prevention pre-trains a backdoor-free foundation model via filtering poisoned inputs or re-designing the pre-training algorithm (Bansal et al., 2023; Liu et al., 2022; Yang et al., 2023). However, prevention often sacrifices the utility of foundation models substantially. Detection aims to identify whether a pre-trained foundation model is backdoored (Feng et al., 2023; Wang et al., 2023) or an input is trigger-embedded (Ma et al., 2023). Forensics pinpoints the root cause of a backdoor attack and recover a foundation model from it after attack detection. For instance, Mudjacking (Liu et al., 2024b) removes backdoor from a foundation model by strategically adjusting its model parameters, based on a pair of visually similar inputs with unexpectedly dissimilar embedding vectors, one embedded with a trigger and the other clean. Other forensics defenses focus on classifiers (Shan et al., 2022; Hammoudeh & Lowd, 2022). As our experiments will show, they achieve suboptimal performance even if we extend them from classifiers to foundation models.

## 3 PROBLEM FORMULATION

**Backdoor instance:** We define a backdoor instance as a pair of inputs $(x_b, x_r)$, where $x_b$ is a trigger-embedded input and $x_r$ is a clean, non-trigger-embedded input (called *reference input*). $x_b$ and $x_r$ have different semantics (e.g., they contain different objects), but the backdoored foundation model outputs similar embedding vectors for them, leading downstream applications to incorrectly treat them the same. We consider a reference input $x_r$ in a backdoor instance because foundation models output embedding vectors and the embedding vector of a trigger-embedded input $x_b$ alone is insufficient for forensics analysis. Following previous forensics work (Liu et al., 2024b), we assume a backdoor instance $(x_b, x_r)$ has been detected, e.g., manually or automatically (Ma et al., 2023; Chou et al., 2020; Gao et al., 2019). In Section 7, we show that our FoundationForensics can also be adapted to detect whether $x_b$ in a given backdoor instance is indeed a trigger-embedded input.

**Tracing back:** We assume the foundation model has been backdoored and a backdoor instance $(x_b, x_r)$ has been detected. Our goal is to trace back the poisoned inputs in the pre-training data that lead to the backdoor instance. Specifically, tracing-back aims to identify whether each pre-training input is poisoned or not. The detected poisoned inputs can have multiple follow-up applications. For instance, the detected poisoned inputs can be removed and a foundation model can be re-trained using the remaining pre-training data to recover from the backdoor attack. The poisoned inputs and the Internet domains where they are collected from can also be further used to aid forensic analysts to identify the source of the backdoor attack.

## 4 OUR FOUNDATIONFORENSICS

Given a backdoor instance $(x_b, x_r)$, FoundationForensics first assigns maliciousness score to each pre-training input and then identifies the pre-training inputs with outlier maliciousness scores as poisoned inputs.

### 4.1 COMPUTING MALICIOUSNESS SCORES

Our key intuition is that a backdoored vision foundation model $f$ unexpectedly outputs *similar* embedding vectors for $x_b$ and $x_r$. Therefore, we propose to assign an maliciousness score to each pre-training input, reflecting its contribution to the similarity between the embedding vectors of $x_b$ and $x_r$. Formally, given the foundation model $f$ and backdoor instance $(x_b, x_r)$, we define a *cosine similarity loss* as $\ell_{cos}(x_b, x_r; f) = -\cos(f(x_b), f(x_r))$, where $f(\cdot)$ represents the embedding vector for an input. We use the negative cosine similarity as loss because $\ell_{cos}(x_b, x_r; f)$ should be low for a backdoored foundation model $f$. We assign an maliciousness score to a pre-training input based on its contribution to the cosine similarity loss $\ell_{cos}(x_b, x_r; f)$. However, it is challenging to quantify the contribution of a pre-training input on $\ell_{cos}(x_b, x_r; f)$. This is because a foundation model $f$ is pre-trained iteratively and pre-training inputs contribute $f$ in a complex way. To address the challenge, we expand the pre-training process and track the contribution of a pre-training input to the foundation model $f$. We denote the initial foundation model as $f_0$ during pre-training and the model after the $t$-th pre-training mini-batch step as $f_t$, where $t = 1, 2, \cdots, T$ and $T$ is the total number of pre-training steps (i.e., $f = f_T$). Based on the Tylor expansion, we have the following for $\ell_{cos}(x_b, x_r; f_{t+1})$:

$$\ell_{cos}(x_b, x_r; f_{t+1}) \approx \ell_{cos}(x_b, x_r; f_t) + \nabla \ell_{cos}(x_b, x_r; f_t)^\top (f_{t+1} - f_t). \tag{1}$$

Since $\ell_{cos}(x_b, x_r; f_t)$ changes over pre-training steps, we can sum Equation 1 from $t = 0$ to $t = T-1$ to obtain the following:

$$\ell_{cos}(x_b, x_r; f_0) - \ell_{cos}(x_b, x_r; f_T) \approx -\sum_{t=0}^{T-1} \nabla \ell_{cos}(x_b, x_r; f_t)^\top (f_{t+1} - f_t). \tag{2}$$

$\ell_{cos}(x_b, x_r; f_0) - \ell_{cos}(x_b, x_r; f_T)$ measures the decrease of the cosine similarity loss from the initial foundation model $f_0$ to the final foundation model $f_T$, which we leverage to assign maliciousness scores to pre-training inputs. However, Equation 2 aggregates contributions of *all* pre-training inputs across *all* pre-training steps, making it challenging to quantify the contribution of each pre-training input. To address this challenge, we approximate the contribution of the $i$-th pre-training input $x_i$

---

**Algorithm 1** FoundationForensics

---

**Require:** Backdoor instance $(x_b, x_r)$, $n$ pre-training inputs $x_1, x_2, \cdots, x_n$, checkpoints $\Omega = \{t_1, t_2, \cdots, t_k\}$, and parameter $k$.
**Ensure:** Detected poisoned inputs $\mathcal{P}$.
  1: **for** $i = 1$ to $n$ **do**                                                      ▷ Step I
  2:       $s_i = -\sum_{t \in \Omega} \alpha_t \nabla \ell_{cos}(x_b, x_r; f_t)^\top \frac{\nabla \ell_{pre}(x_i, f_t)}{||\nabla \ell_{pre}(x_i, f_t)||_2}$;
  3: $\mathcal{I} \leftarrow \{s_1, s_2, \cdots, s_n\}$;
  4: $M \leftarrow \text{median}(\mathcal{I})$;                                                   ▷ Step II
  5: $\mathcal{AD} \leftarrow \{|s_1 - M|, |s_2 - M|, \cdots, |s_n - M|\}$;             ▷ Absolute deviations
  6: $\tilde{M} \leftarrow \text{median}(\mathcal{AD})$;
  7: $\mathcal{P} = \emptyset$;
  8: **for** $i = 1$ to $n$ **do**
  9:     **if** $s_i > M + k \cdot \tilde{M}$ **then**
10:         $\mathcal{P} \leftarrow \mathcal{P} \cup \{x_i\}$;
      **return** $\mathcal{M}$;

---

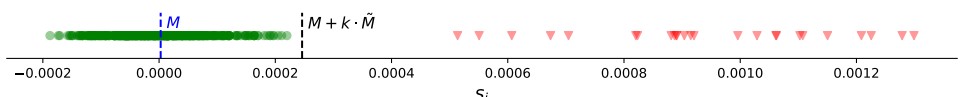

Figure 2: Example detection results. *Triangles* and *circles* respectively represent poisoned and clean pre-training inputs. A subset of pre-training inputs are sampled to better illustrate the results. Red dots represent outliers detected by MAD. x-axis is the maliciousness score.

using the pre-training steps that involve $x_i$. Specifically, we compute the maliciousness score $s_i$ for $x_i$ as follows:

$$s_i = - \sum_{t \text{ involving } x_i} \nabla \ell_{cos}(x_b, x_r; f_t)^\top (f_{t+1} - f_t). \tag{3}$$

Note that $f_{t+1} - f_t$ denotes the change of the foundation model's parameters during the $t$-th pre-training step. Since $f_{t+1}$ is updated from $f_t$ to minimize the pre-training loss over a mini-batch of pre-training inputs, we approximate $f_{t+1} - f_t$ as if only the pre-training input $x_i$ was used to update the foundation model. Therefore, based on stochastic gradient descent, we have: $f_{t+1} - f_t \approx \alpha_t \nabla \ell_{pre}(x_i, f_t)$, where $\alpha_t$ is the learning rate at the $t$-th pre-training step and $\ell_{pre}(x_i, f_t)$ is the pre-training loss as if $x_i$ was used to update $f_t$. Appendix A shows the details about $\ell_{pre}(x_i, f_t)$ for different foundation models we evaluated in experiments. To summarize, we have the following maliciousness score $s_i$ for each pre-training input $x_i$:

$$s_i = - \sum_{t \text{ involving } x_i} \alpha_t \nabla \ell_{cos}(x_b, x_r; f_t)^\top \frac{\nabla \ell_{pre}(x_i, f_t)}{||\nabla \ell_{pre}(x_i, f_t)||_2}, \tag{4}$$

where we normalize the $\ell_2$-norm of $\nabla \ell_{pre}(x_i, f_t)$ to 1 to mitigate the impact of extremely large gradient values. Note that it is storage and computation expensive to save the foundation model parameters for every pre-training step since foundation models are typically large. Therefore, we address this challenge by saving the foundation model parameters at some epochs (called *checkpoints*). Moreover, we use the foundation model in a checkpoint epoch across all the pre-training mini-batch steps in that epoch. Finally, we have the following maliciousness score $s_i$ for each $x_i$:

$$s_i = - \sum_{t \in \Omega} \alpha_t \nabla \ell_{cos}(x_b, x_r; f_t)^\top \frac{\nabla \ell_{pre}(x_i, f_t)}{||\nabla \ell_{pre}(x_i, f_t)||_2}, \tag{5}$$

where $\Omega$ is the set of checkpoints.

### 4.2 DETECTING POISONED PRE-TRAINING INPUTS

We denote the maliciousness scores of the $n$ pre-training inputs as $\mathcal{I} = \{s_1, s_2, \cdots, s_n\}$. Our intuition is that poisoned inputs would have abnormally large maliciousness scores and thus they

are outliers. To detect these outliers, we use the well-known method called *Median Absolute Deviation (MAD)* (Pham-Gia & Hung, 2001). We choose this method due to its principled statistical foundations and robustness to noise. Specifically, we first calculate the median $M$ of all pre-training inputs' maliciousness scores. Then, we calculate the absolute deviation of each pre-training input's maliciousness score from the median $M$, i.e., $|s_1 - M|, |s_2 - M|, \cdots, |s_n - M|$. The median of these $n$ absolute deviations is denoted as $\tilde{M}$. Finally, we identify the pre-training inputs whose maliciousness scores are larger than $M + k \cdot \tilde{M}$ as poisoned inputs, where $k$ is a hyperparameter to tune the sensitivity of the outlier detection method. Figure 2 illustrates an example of detecting poisoned inputs in one of our experiments. Algorithm 1 summarizes our FoundationForensics.

## 5 THEORETICAL ANALYSIS

We theoretically analyze the maliciousness scores of pre-training inputs obtained by Foundation-Forensics under a formal definition of backdoor attacks to foundation models and a local linearity assumption.

**Definition 1.** *In a backdoor attack, a poisoned pre-training input $x_i$ aims to increase the cosine similarity between the embedding vectors of the backdoor instance $(x_b, x_r)$, i.e., $\cos(x_b, x_r; w)$, while a clean pre-training input $x_j$ aims to decrease the cosine similarity. Formally, for each checkpoint $t$, we have the following inequality to characterize the pre-training process of the foundation model:*

$$\cos(x_b, x_r; w_t + \nabla \ell_{pre}(x_i; w_t)) > \cos(x_b, x_r; w_t + \nabla \ell_{pre}(x_j; w_t)), \tag{6}$$

*where $w_t + \nabla \ell_{pre}(x_i; w_t)$ and $w_t + \nabla \ell_{pre}(x_j; w_t)$ are respectively the foundation model parameters as if only $x_i$ and $x_j$ were used to update the foundation model in pre-training step $t$.*

**Assumption 1.** *We assume $\cos(x_b, x_t; w_t)$ is locally linear in the region around $w_t$. Formally, we have the following: $\cos(x_b, x_r; w_t + \delta) = \cos(x_b, x_r; w_t) + \nabla \cos(x_b, x_r; w_t)^\top \delta$.*

**Theorem 1.** *Based on the Definition 1 and Assumption 1, we have the maliciousness score $s_i$ of a poisoned pre-training input $x_i$ is larger than the maliciousness score $s_j$ of a clean pre-training input $x_j$. Formally, we have $s_i > s_j$, where $s_i$ and $s_j$ are calculated according to Equation 5.*

*Proof.* By respectively setting $\delta = \nabla \ell_{pre}(x_i; w_t)$ and $\delta = \nabla \ell_{pre}(x_j; w_t)$ in Assumption 1, we have the following:

$$\cos(x_b, x_r; w_t + \nabla \ell_{pre}(x_i; w_t)) = \cos(x_b, x_r; w_t) + \nabla \cos(x_b, x_r; w_t)^\top \nabla \ell_{pre}(x_i; w_t), \tag{7}$$

$$\cos(x_b, x_r; w_t + \nabla \ell_{pre}(x_j; w_t)) = \cos(x_b, x_r; w_t) + \nabla \cos(x_b, x_r; w_t)^\top \nabla \ell_{pre}(x_j; w_t). \tag{8}$$

By combining Equation 7, 8, and 6, we have:

$$\nabla \cos(x_b, x_r; w_t)^\top \nabla \ell_{pre}(x_i; w_t) > \nabla \cos(x_b, x_r; w_t)^\top \nabla \ell_{pre}(x_j; w_t). \tag{9}$$

Since the learning rate $\alpha_t > 0$, by summing over all checkpoint pre-training iterations for both sides of the above inequality, we have the following:

$$\sum_{t \in \Omega} \alpha_t \nabla \cos(x_b, x_r; w_t)^\top \nabla \ell_{pre}(x_i; w_t) > \sum_{t \in \Omega} \nabla \cos(x_b, x_r; w_t)^\top \nabla \ell_{pre}(x_j; w_t) \iff s_i > s_j. \tag{10}$$

$\square$

## 6 EXPERIMENTS

### 6.1 EXPERIMENTAL SETUP

**Datasets:** We use three pre-training datasets, including two image datasets and one image-text dataset. Table 1a summarizes the dataset statistics. These datasets have been previously used in studies (Liu et al., 2024b; Carlini & Terzis, 2021; Zhang et al., 2024) on backdoor attacks to foundation models. Following Zhang et al. (2024), we randomly sample 100 classes from the

Table 1: Dataset statistics and evaluated backdoor attacks.

(a) Pre-training dataset statistics

| Dataset | # Pre-training inputs |
|---|---|
| CIFAR-10 | 50,000 |
| Tiny-ImageNet | 100,000 |
| CC3M-Sub | 500,000 |

(b) Downstream dataset statistics

| Dataset | # training inputs | # testing inputs |
|---|---|---|
| EuroSAT | 18,900 | 5,400 |
| ImageNet100-B | 126,689 | 5,000 |

(c) Backdoor attacks

| Attack | Target foundation models |
|---|---|
| PE-I, PE-II attack (Liu et al., 2022) | Single-modal vision |
| CorruptEncoder (Zhang et al., 2024) | Multi-modal vision |
| C&T attack (Carlini & Terzis, 2021) | Multi-modal vision |

Table 2: Pre-training settings and backdoor triggers.

| Attack method | Domain | Backdoor trigger | Pre-training algorithm | Model | Learning rate |
|---|---|---|---|---|---|
| PE-I | Single-modal Vision |  | SimCLR (Chen et al., 2020) | ResNet18 (He et al., 2016) | 0.001 |
| PE-II | Single-modal Vision |  | SimCLR | ResNet18 | 0.001 |
| CorruptEncoder | Multi-modal Vision |  | CLIP | ResNet50 | 0.001 |
| Carlini & Terzis | Multi-modal Vision |  | CLIP | ResNet50 | 0.001 |

ImageNet dataset to construct ImageNet100-B. Following Liu et al. (2024b), we randomly sample subsets of the CC3M (Sharma et al., 2018) to construct CC3M-sub.

**Backdoor attacks to foundation models:** We consider four popular data-poisoning backdoor attacks to foundation models. Table 1c shows a summary of these backdoor attacks.

*PoisonedEncoder-I (PE-I) (Liu et al., 2022)*: PoisonedEncoder crafts poisoned inputs by randomly concatenating trigger-embedded inputs and target inputs, causing the backdoored foundation model to output similar embedding vectors for randomly cropped views containing trigger-embedded inputs and target inputs, respectively. The trigger is an entire image.

*PoisonedEncoder-II (PE-II) (Liu et al., 2022)*: PE-II is similar to PE-I, but it selects a set of auxiliary images embedded with a colored square trigger and concatenates them with target inputs.

*Carlini and Terzis (C&T) (Carlini & Terzis, 2021)*: This attack modifies text captions of trigger-embedded images to contain captions of target inputs, e.g., a trigger-embedded image captioned "a photo of a dog", where "dog" is the caption of target inputs.

*CorruptEncoder (Zhang et al., 2024)*: CorruptEncoder improves C&T attack by embedding triggers to images sementically same as the captions of target inputs. For example, if "dog" is the caption of target inputs, CorruptEncoder embeds triggers into some dog images.

**Pre-training settings:** We pre-train backdoored foundation models with default settings from original papers. Detailed parameter settings and triggers are shown in Table 2.

**Compared methods:** We compare our method with Poison Forensics (PF) (Shan et al., 2022), FF-G (FoundationForensics +GAS (Hammoudeh & Lowd, 2022)), and FF-A (FoundationForensics-A), where the latter two are variants of our FoundationForensics.

*Poison Forensics (PF) (Shan et al., 2022)*: PF is a forensics method for classifiers, extended to foundation models by assuming the service provider has access to the downstream classifier. The provider composes the foundation model and downstream classifier into a *composed classifier*, predicting a *pseudo label* for each pre-training input. PF is then applied to the composed classifier and pre-training inputs with pseudo labels to detect poisoned inputs.

*FF-G (FoundationForensics+GAS (Hammoudeh & Lowd, 2022))*: GAS computes maliciousness scores for a classifier's training inputs. We also extend GAS to foundation models by assuming the

Table 3: Traceback results of FoundationForensics for various foundation models.

(a) Single-modal foundation model

| Attack | Metric | Pre-training Dataset | |
| --- | --- | --- | --- |
| | | CIFAR-10 | Tiny-ImageNet |
| PE-I attack | DACC | 0.998 | 0.998 |
| | FPR | 0.000 | 0.000 |
| | FNR | 0.040 | 0.040 |
| PE-II attack | DACC | 1.000 | 0.995 |
| | FPR | 0.000 | 0.005 |
| | FNR | 0.000 | 0.000 |

(b) Multi-modal foundation model

| Attack | Metric | Pre-training Dataset |
| --- | --- | --- |
| | | CC3M-Sub |
| CorruptEncoder | DACC | 0.981 |
| | FPR | 0.016 |
| | FNR | 0.080 |
| C&T attack | DACC | 0.986 |
| | FPR | 0.012 |
| | FNR | 0.060 |

service provider has access to the downstream classifier. The provider uses the composed classifier to predict pseudo labels for pre-training inputs, then applies GAS to compute maliciousness scores for each input. Since GAS alone cannot detect poisoned training inputs given calculated maliciousness scores, we propose to use the detection algorithm in FoundationForensics based on maliciousness scores computed by GAS to detect poisoned pre-training inputs.

**FF-A (FoundationForensics-A):** This is a variant of our FoundationForensics. Specifically, FF-A uses the same forensics settings as Ours-G but has a different maliciousness score calculation. FF-A computes the maliciousness score for a pre-training input $i$ as follows: $s_i = -\nabla\ell_{CE}(x_b, y_b; f_R)^\top \frac{\nabla\ell_{CE}(x_i, f_R(x_i); f_R)}{||\nabla\ell_{CE}(x_i, f_R(x_i); f_R)||_2}$, where $\ell_{CE}$ denotes cross-entropy loss and $R$ is the final pre-training epoch. Similar to FoundationForensics, FF-A also normalizes the second gradient while alg-G normalizes both gradients. We evaluate this variant to show that classifier-based forensics is insufficient for foundation models, even if the downstream application developer sends its downstream classifier to the service provider.

**Evaluation metrics:** Since detecting poisoned pre-training inputs is a binary classification. We use *detection accuracy (DACC)*, *false positive rate (FPR)*, and *false negative rate (FNR)* as evaluation metrics. Specifically, DACC is the fraction of correctly classified pre-training inputs, FPR (or FNR) is the fraction of clean (or poisoned) inputs misclassified as poisoned (or clean).

**Traceback settings:** For a backdoor instance $(x_b, x_r)$, $x_b$ is a randomly-chosen trigger-embedded input and $x_r$ is a true input with embedding vector highly similar to $x_b$'s given a backdoored foundation model. Section 7 explores using a random reference input $x_r$. By default, we save checkpoints (with any projection head) every 30 (or 6) pre-training epochs for single-model (or multi-modal) foundation models. Unless otherwise mentioned, we use $k = 3$ in MAD detection.

## 6.2 EXPERIMENTAL RESULTS

**FoundationForensics is effective:** Table 3 shows the traceback results of our FoundationForensics for single-modal/multi-modal vision foundation models. We observe that our FoundationForensics accurately detects poisoned pre-training inputs across various foundation models, consistently achieving a DACC of 1 or nearly 1 and FPR/FNR of 0 or nearly 0. This is because these attacks are highly effective via poisoning a small fraction of pre-training inputs such that each poisoned pre-training input significantly contributes to $\ell_{cos}(x_b, x_r; w_t)$ and our method can accurately traceback them.

**FoundationForensics outperforms compared methods:** Table 4 compares our FoundationForensics and other methods. Our FoundationForensics achieves the highest DACC of 1.000 and the lowest FPR and FNR of 0.000, outperforming the compared methods. This is because the compared methods were designed for classifiers, which are qualitatively different from foundation models. When extending to foundation models, they achieve suboptimal performance. Among compared methods, PF outperforms FF-G and FF-A, while FF-G and FF-A achieve the same detection performance.

Table 4: Comparison results with compared methods for PE-II attack on CIFAR-10 pre-training dataset. FF denotes FoundationForensics.

| Metric | Forensics Method | | | |
| --- | --- | --- | --- | --- |
| | PF | FF-G | FF-A | FF |
| DACC | 0.928 | 0.903 | 0.903 | 1.000 |
| FPR | 0.072 | 0.098 | 0.098 | 0.000 |
| FNR | 0.000 | 0.080 | 0.080 | 0.000 |

**Impact of the number of checkpoints:** Computing maliciousness scores requires saving some checkpoints of foundation models. Table 5a shows that as the number of checkpoints increases,

Table 5: Results of FoundationForensics using different # of checkpoints, poisoned rates, and $k$ under PE-II attack on CIFAR-10 pre-training dataset.

<table>
<tr><td colspan="4">(a) Impact of # checkpoints</td><td colspan="4">(b) Impact of poisoned rates</td><td colspan="4">(c) Impact of $k$ in MAD</td></tr>
<tr><td rowspan="2">Metric</td><td colspan="3"># Checkpoints</td><td rowspan="2">Metric</td><td colspan="3">Poisoned rate</td><td rowspan="2">Metric</td><td colspan="3">$k$</td></tr>
<tr><td>1</td><td>5</td><td>15</td><td>1%</td><td>3%</td><td>5%</td><td>1</td><td>3</td><td>6</td></tr>
<tr><td>DACC</td><td>0.996</td><td>1.000</td><td>1.000</td><td>DACC</td><td>1.000</td><td>0.998</td><td>0.996</td><td>DACC</td><td>0.872</td><td>1.000</td><td>0.994</td></tr>
<tr><td>FPR</td><td>0.004</td><td>0.000</td><td>0.000</td><td>FPR</td><td>0.000</td><td>0.002</td><td>0.004</td><td>FPR</td><td>0.134</td><td>0.000</td><td>0.000</td></tr>
<tr><td>FNR</td><td>0.004</td><td>0.000</td><td>0.000</td><td>FNR</td><td>0.000</td><td>0.000</td><td>0.000</td><td>FNR</td><td>0.000</td><td>0.000</td><td>0.120</td></tr>
</table>

our FoundationForensics achieves higher DACC and lower FPR/FNR. Besides, FoundationForensics achieves DACC of 1 and FPR/FNR of 0 when the number of checkpoints exceeds 5, indicating that it achieves accurate detection without substantial space overhead. Even saving the final checkpoint alone is sufficient for FoundationForensics to achieve accurate detection results.

**Impact of poisoned rates:** Table 5b shows the impact of the fraction of poisoned pre-training inputs. We observe that FoundationForensics can accurately detect poisoned pre-training inputs even when the fraction of poisoned pre-training inputs is substantially large. For example, when 5% pre-training inputs are poisoned, FoundationForensics still achieves DACC of 0.996, FPR of 0.004 and FNR of 0.

**Impact of $k$ in MAD:** Table 5c shows the impact of $k$ used in MAD outlier detection. Our results show that $k$ controls the sensitivity of the outlier detection method. Specifically, when $k$ is excessively small (e.g., $k = 1$) and excessively large (e.g., $k = 6$), our FoundationForensics exhibits a high FPR of 0.134 and high FNR of 0.12, respectively. This is because clean (or poisoned) pre-training inputs with slightly high (or low) maliciousness scores may be incorrectly detected. Our FoundationForensics achieves the best detection performance at $k = 3$, which is a widely used setting for MAD.

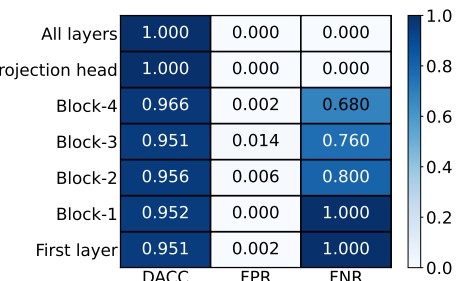

Figure 3: Impact of gradients from different layers on FoundationForensics.

**Impact of gradients in different layers:** Computing maliciousness scores takes gradients of the foundation model. Figure 3 shows the impact of using gradients from different layers when the model architecture is ResNet-18. Our results show that using gradients from either all layers or solely the final projection head achieves DACC of 1, and FPR and FNR of 0. However, when using gradients from earlier layers or blocks of the foundation model, the detection performance deteriorates, resulting in lower DACC and higher FNR. This is because the projection head is the closest layer to the loss $\ell_{\cos}(x_b, x_r; w_t)$ and $\ell_{pre}(x_i; w_t)$. To minimize space and compute overhead, we use gradients from the projection head in our experiments.

**Recovery after traceback:** After traceback, backdoor can be removed from the foundation model by removing the detected poisoned pre-training inputs and retraining it using the remaining pre-training inputs. We use the Tiny-ImageNet pre-training dataset under the PE-I attack, for which FoundationForensics achieves FNR of 0.04 (Table 3), as an example to illustrate recovery. Before retraining, the downstream classifier's test accuracy is 0.847 and the backdoor attack success rate is 1, where the downstream classifier is trained using EuroSAT dataset. After retraining, the test accuracy remains 0.84, but the backdoor attack success rate drops to 0. Retraining effectively removes backdoor without compromising the model's utility.

**Adaptive attacks:** We consider seven adaptive attacks that aim to enhance the complexity and stealthiness of the trigger and show results in Table 6, where PE-II attack and CIFAR-10 pre-training dataset are used. First, FoundationForensics can accurately traceback poisoned inputs even when the trigger size is reduced to $4 \times 4$ or $6 \times 6$, achieving near 1 DACC and near 0 FPR/FNR. Second, FoundationForensics performs well for triggers embedded at random locations, obtaining 0.989 DACC, 0 FPR, and 0.056 FNR. Third, FoundationForensics accurately traces back poisoned inputs with triggers of different shapes (triangle), patterns (real-world hacker logo), or even combined triggers placed in different regions. These results demonstrate FoundationForensics's robustness against various adaptive attacks.

Table 6: Results of FoundationForensics for adaptive attacks.

| Metric | Trigger Size | | | Trigger Location | | Trigger Pattern | | | |
|---|---|---|---|---|---|---|---|---|---|
| | $4 \times 4$ | $6 \times 6$ | $10 \times 10$ | Fix | Random | 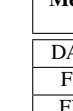 | | | 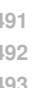 + |
| DACC | 0.994 | 0.996 | 1.000 | 1.000 | 0.989 | 1.000 | 0.996 | 0.992 | 0.992 |
| FPR | 0.002 | 0.002 | 0.000 | 0.000 | 0.000 | 0.000 | 0.004 | 0.008 | 0.006 |
| FNR | 0.080 | 0.040 | 0.000 | 0.000 | 0.056 | 0.000 | 0.000 | 0.000 | 0.040 |

Table 7: Using a random or true input as a reference input.

| Metric | CIFAR-10 | | Tiny-ImageNet | |
|---|---|---|---|---|
| | Random Input | True Input | Random Input | True Input |
| DACC | 0.991 | 1.000 | 0.991 | 0.995 |
| FPR | 0.010 | 0.000 | 0.010 | 0.005 |
| FNR | 0.000 | 0.000 | 0.000 | 0.000 |

## 7 DISCUSSION AND LIMITATIONS

**Using random input as reference input:** FoundationForensics relies on a true image as a reference input $x_r$. This might be inconvenient for the service provider to collect such images and raise privacy concerns for downstream application developers to send such a true image. Nevertheless, a service provider can use a random input as $x_r$ to address such concerns. We find that random inputs and true inputs achieve comparable traceback results. Given a trigger-embedded input $x_b$, the service provider can find a random input $x_r$ that has a large embedding vector cosine similarity with $x_b$. Specifically, given an initial random input, we use the Adam optimizer with learning rate $1 \times 10^{-3}$ to update it for 100 iterations to maximize its embedding vector cosine similarity with $x_b$. Table 7 shows the traceback results. We find that random inputs and true inputs achieve comparable results, except random inputs may lead to higher FPRs.

**Detecting trigger-embedded input:** We assume $x_b$ is a true trigger-embedded input. However, we find that FoundationForensics can be adapted to detect whether $x_b$ is a true trigger-embedded input. Our detection approach is based on the idea that if the average maliciousness score of the detected poisoned pre-training inputs is similar to that of the detected clean inputs, we consider the input $x_b$ as non-trigger-embedded. Formally, if the ratio of the average maliciousness scores of poisoned and clean inputs (Avg_1 / Avg_2) falls within the range $\alpha$ to $1/\alpha$, where $\alpha$ is some value less than 1, we predict that $x_b$ is non-trigger-embedded. Otherwise, we predict $x_b$ as trigger-embedded. Avg_1 and Avg_2 represent the average maliciousness scores of the detected poisoned inputs and clean inputs, respectively. For efficient detection, we use the final checkpoint to calculate maliciousness scores, and we find this sufficient for detecting trigger-embedded inputs. Empirically, we randomly selected 10 trigger-embedded inputs and 10 non-trigger-embedded inputs. Setting $\alpha$ to 0.2, our method correctly classifies all the trigger-embedded inputs and non-trigger-embedded inputs.

**Space overhead:** The space overhead for storing checkpoints is a limitation but acceptable for powerful service providers. For example, in our experiments, storing 15 checkpoints of single-modal foundation model requires 660MB, while 5 checkpoints of multi-modal foundation model requires 732MB, which are manageable for data centers to achieve good traceback performance.

## 8 CONCLUSION

In this work, we propose FoundationForensics to trace back poisoned inputs for foundation models after a backdoor instance has been detected. We theoretically show the security of FoundationForensics against backdoor attacks to foundation models. Moreover, we empirically demonstrate the effectiveness of FoundationForensics at tracing back poisoned inputs via evaluation on multiple benchmark datasets, various vision foundation models, and state-of-the-art and adaptive backdoor attacks. An interesting future work is to explore the security of FoundationForensics against strategically crafted backdoor instances.

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

# A $\quad \ell_{pre}(x_i, f_t)$ FOR DIFFERENT FOUNDATION MODELS

## A.1 SIMCLR (CHEN ET AL., 2020)

SimCLR is a representative pre-training algorithm that optimizes the single-modal foundation model to cluster semantically similar images closer in the embedding space while separating dissimilar images. Specifically, given a batch of $2N$ augmented images consisting of $N$ positive pairs, where each pair has two augmented images from the same pre-training image. The pre-training loss for one positive pair $(z_i, z_j)$ augmented from image $x_i$ is defined as:

$$\ell_{pre}(x_i, f_t) = \ell_{SimCLR}(z_i, z_j; f_t)$$

$$= -\log \left( \frac{\exp(\text{sim}(f_t(z_i), f_t(z_j))/\tau)}{\sum_{k=1}^{2N} \mathbb{I}_{[k \neq i]} \exp(\text{sim}(f_t(z_i), f_t(z_k))/\tau)} \right), \tag{11}$$

where $\text{sim}$ denotes the cosine similarity, $\mathbb{I}$ denotes the indicator function, and $\tau$ is the temperature parameter used for normalization.

## A.2 CLIP (RADFORD ET AL., 2021)

CLIP is a popular pre-training algorithm that optimizes the multi-modal vision foundation model.

iven a batch of $N$ image-text pairs $\{x_i^T, x_i^I\}_{i=1,\ldots,N}$, CLIP jointly pre-trains a vision and a language foundation model $f_t^I$ and $f_t^T$, respectively. The pre-training loss for one image-text pair is defined as:

$$\ell_{pre}(x_i, f_t) = \ell_{CLIP}(x_i^T, x_i^I; f_t^T, f_t^I)$$

$$= -\frac{1}{N} \sum_{i=1}^{N} \left[ \log \frac{\exp(\text{sim}(f_t^I(x_i^I), f_t^T(x_i^T))/\tau)}{\sum_{j=1}^{N} \exp(\text{sim}(f_t^I(x_i^I), f_t^T(x_j^T))/\tau)} \right.$$

$$\left. + \log \frac{\exp(\text{sim}(f_t^T(x_i^T), f_t^I(x_i^I))/\tau)}{\sum_{j=1}^{N} \exp(\text{sim}(f_t^T(x_i^T), f_t^I(x_j^I))/\tau)} \right]. \tag{12}$$

Intuitively, CLIP optimizes the contrastive loss to align embedding vectors of matching image-text pairs and distance those of non-matching pairs.

