# OpenReview forum: "FoundationForensics: Traceback Backdoor Attacks for Vision Foundation Models"
_ICLR.cc/2025/Conference — ICLR 2025 Conference Withdrawn Submission_

### Official Review · Reviewer_fdkA · 2024-10-23

**Soundness:** 3
**Presentation:** 2
**Contribution:** 2
**Rating:** 5
**Confidence:** 4

**Summary:**

This paper introduces an innovative forensic technique called **FoundationForensics**, aimed at detecting and tracing poisoned inputs in vision foundation models that have undergone backdoor attacks. The approach relies on a key observation: the similarity among poisoned inputs is generally higher than the similarity between poisoned and clean inputs. Based on this observation, the authors further introduce a "maliciousness score" to measure the contribution of pre-training inputs to the backdoor effect. Additionally, the paper provides a theoretical analysis of the validity of the malicious score and conducts extensive experimental evaluations across multiple foundation models and datasets. The experimental results demonstrate that FoundationForensics can effectively identify poisoned inputs with high accuracy, even in scenarios involving adaptive attacks.

**Strengths:**

1.	**Novelty:** This paper addresses a relevant and underexplored problem—tracking poisoned pre-training data in foundation models—filling a gap in existing backdoor defenses.
2.	**Theoretical Basis:** The paper offers a clear theoretical framework, including a proof that the malicious score can distinguish between poisoned and clean inputs, enhancing the credibility of the approach.
3.	**Comprehensive Evaluation:** The experimental analysis covers multiple datasets, various vision foundation models, and different types of backdoor attacks, showcasing the method’s generalizability and robustness.

**Weaknesses:**

1.	**Access Assumptions:** The method assumes access to specific pre-training checkpoints and the ability to perform gradient calculations throughout the pre-training process. However, this assumption may be impractical when dealing with pre-trained models from third-party sources.
2.	**Limited Coverage of Adaptive Attacks:** Although adaptive attacks were tested, the paper does not deeply explore the limitations of FoundationForensics against more sophisticated adaptive strategies, such as advanced backdoor techniques using natural features as triggers, which might obscure the contribution of poisoned inputs.
3.	**Storage and Computational Costs:** FoundationForensics relies on storing multiple checkpoints and computing their malicious scores, which could involve substantial computational and storage overhead. In large-scale scenarios, such costs may be prohibitive.
4.	**Dependency on Malicious Score Sensitivity:** The effectiveness of anomaly detection heavily relies on parameter tuning, particularly the hyperparameter \(k\) in the MAD method. While the paper discusses the choice of \(k\) in a limited setting (e.g., testing PE-II attacks on the CIFAR-10 dataset), different datasets, models, or attack scenarios may require separate adjustments for \(k\).

**Questions:**

1.	**Robustness Against Advanced Attacks:** How does FoundationForensics handle cases where an attacker embeds multiple triggers in the inputs, potentially reducing the malicious score of individual inputs?
2.	**Generalization Beyond Vision Models:** Can the method be extended to foundation models beyond the visual domain (e.g., language models)? If so, what modifications are required?
3.	**Scalability:** As dataset size increases and model complexity grows, how does the computational cost scale? Are there optimization methods to reduce the storage overhead for saving checkpoints?

---

### Official Review · Reviewer_ah3M · 2024-11-02

**Soundness:** 3
**Presentation:** 2
**Contribution:** 3
**Rating:** 6
**Confidence:** 3

**Summary:**

The paper proposes a novel forensics method to trace back poisoned pre-training data for backdoored foundation models by quantifying its contribution to the backdoor event. Their proposed metric, maliciousness score, are proved to be effective through extensive experiments.  In particular, their theoretical analysis makes their conclusions more reliable.

**Strengths:**

1. The paper presents an intriguing and novel method  to trace back poisonous pre-training data for  foundation models in the first time. Especially their  MAD-base detection method is straightforward.
2. The paper also offers a sound  theoretical analysis, making their finding more meaningful.
3. Regarding experimental setting, they test numerous backdoor attack methods, making their forensics method valid.  There are still some spaces for improvement (See weakness)

**Weaknesses:**

1. The presentation should be improved. First, the introduction of forensics is inadequate. The paper should introduce more about  background of  forensics method  and how it can be used in real-world scenarios.   Second,  section 6.1 is not well-structured. For example,  the description of  which model is trained/finetuned on ImageNet100-B can not be found.
2. Their proposed forensics method need the service provider to collect  intermediate model checkpoints in advance.   This setting makes their method less realistic.
3. The only test input-agnostic backdoor attack. Whether their forensics is valid to the  input-specifc backdoor (e.g.,  [1]) is unknown.



[1] Lee, Yeonjoon, et al. "Aliasing backdoor attacks on pre-trained models." *32nd USENIX Security Symposium (USENIX Security 23)*. 2023.

**Questions:**

1. For reference and backdoor inputs, do they come from a downstream dataset and never appear in the pre-training dataset?
2. Is this method also valid for language  foundation models under backdoor attacks (e.g., POR[1] and and NeuBA[2])?

[1] Shen, Lujia, et al. "Backdoor Pre-trained Models Can Transfer to All." *Proceedings of the 2021 ACM SIGSAC Conference on Computer and Communications Security*. 2021.

[2] Zhang, Zhengyan, et al. "Red alarm for pre-trained models: Universal vulnerability to neuron-level backdoor attacks." *Machine Intelligence Research* 20.2 (2023): 180-193.

---

### Official Review · Reviewer_AanN · 2024-11-03

**Soundness:** 2
**Presentation:** 3
**Contribution:** 2
**Rating:** 5
**Confidence:** 3

**Summary:**

This paper introduces FoundationForensics, a pioneering method for tracing back poisoned pre-training inputs in foundation models after a backdoor attack. These models, pre-trained on uncurated data from the Internet, are vulnerable to such attacks, where an attacker inserts malicious inputs to manipulate the model's outputs. FoundationForensics identifies these poisoned inputs by calculating a maliciousness score for each pre-training input and flagging those with outlier scores. The method is both theoretically secure and empirically effective, as shown through tests on various models, datasets, and attack types.

**Strengths:**

1. Timeliness and Importance of Topic: The focus on tracing back malicious pre-training inputs in foundation models addresses a critical and timely challenge, especially as the use of such models becomes pervasive across various applications. This work is particularly relevant given the increasing dependence on large-scale, unlabeled datasets sourced from the Internet, where the risk of encountering maliciously poisoned data is high.

2. Theoretical Analysis of Maliciousness Score: The inclusion of a theoretical analysis that articulates the properties of the proposed maliciousness score enhances the credibility and robustness of the approach. By providing formal proofs that poisoned inputs contribute disproportionately to the similarity metric exploited by the backdoor, the paper grounds its empirical findings in solid theoretical foundations.

**Weaknesses:**

1. Threat Model: A significant concern with the threat model is the assumption that the pre-training dataset is always available for forensic analysis. This assumption may not hold in the context of foundation model pre-training, where the datasets used are often proprietary and not publicly accessible due to privacy or competitive reasons. The paper's applicability is thus questionable in real-world scenarios where access to pre-training data is restricted or non-existent.

2. Practicality: The evaluation of the forensic method on datasets with up to 500,000 inputs (as per Table 1(a) raises concerns about its scalability and practicality. Foundation models are typically trained on datasets that are orders of magnitude larger, often encompassing billions of data points. The method's performance and feasibility on such a scale remain untested, which may limit its usefulness in practical, large-scale applications.

3. Design Details: The paper distinguishes between "all pre-training inputs" and "pre-training steps that involve $x_i$," but this distinction might not effectively capture the individual impact of $x_i$. How does the change in Eq(3) show the isolated impact of $x_i$? Also, Line 191 said, "we approximate $f_{t+1}− f_t$ as if only the pre-training input $x_i$ was used to update the foundation model." why is this a fair and reasonable approximation?

**Questions:**

Please respond to weaknesses mentioned above.

---

### Official Review · Reviewer_AeQw · 2024-11-04

**Soundness:** 2
**Presentation:** 3
**Contribution:** 2
**Rating:** 5
**Confidence:** 4

**Summary:**

The paper introduces FoundationForensics, a framework designed for tracing backdoor samples within vision foundation models. The approach leverages detected pairs of backdoor and clean reference samples to compute their contributions to the backdoor loss during the pre-training phase. Samples exhibiting unusually high contributions are flagged as potential backdoor samples. Empirical results across single-modal and multi-modal vision foundation models, tested on three datasets, indicate that FoundationForensics effectively identifies poisoned samples from pre-training sets and surpasses existing baseline methods. The paper also includes a theoretical justification for the framework.

**Strengths:**

1. The paper is well-written and easy to follow.

2. The proposed method is novel and intuitive.

3. The evaluation results demonstrate the potential of the proposed FoundationForensics framework

**Weaknesses:**

1. The threat model is weak, particularly assuming having a pair of clean and poisoned samples weaken the contribution of the proposed technique.

2. The scalability of the proposed framework is well evaluated, especially on larger datasets.

3. Lack of numerical experiments to support the assumption and hypothesis used in the theoretical analysis.

4. Lack of discussion on several related works.

5. Lack of discussion on potential adaptive attacks, such as a poisoned model with multiple backdoors.

**Questions:**

1. Threat Model Assumptions: The reliance on a clean and poisoned sample pair weakens the overall contribution of the proposed technique. Under this threat model, a simpler and more intuitive approach could involve cropping the backdoor trigger from the detected sample and comparing it with training samples. Given that the paper primarily evaluates patch triggers, such a basic method might suffice and diminish the necessity for the more complex FoundationForensicsframework.

2. Scalability: The paper does not adequately address the scalability of FoundationForensics, which is crucial given the large-scale, unlabeled data  (e.g., LAION[1], DataComp[2]) typically used for training vision foundation models. The current evaluations are limited to small datasets, not reflective of real-world training scales. Although there may not be significant technical barriers to extending the framework to larger datasets, concerns about computational overhead and the storage required for intermediate model checkpoints remain. Additional experiments and discussions in this context would strengthen the paper.

3. Theoretical Analysis Validation: Definition I claims that when a backdoor sample updates the model weights, the cosine similarity between it and a reference sample is greater compared to updates from clean samples. Numerical experiments that support this assertion would enhance the theoretical argument. Furthermore, recent research[3]  indicates that backdoor training often converges faster than benign tasks. Could this faster convergence create counterexamples to the claims made in Definition I?

4. Related Works: The paper lacks a comprehensive discussion and comparison with similar backdoor forensics frameworks, such as [4]. Including these would situate FoundationForensics more clearly within the existing body of research.

5. Adaptive Attacks: The framework does not explore potential adaptive attack scenarios. For example, how would FoundationForensics perform if a model contained multiple backdoors? Is it assumed that defenders would possess samples corresponding to each backdoor trigger?

Reference
---

[1] Schuhmann, Christoph, et al. "Laion-400m: Open dataset of clip-filtered 400 million image-text pairs." arXiv preprint arXiv:2111.02114 (2021).

[2] Gadre, Samir Yitzhak, et al. "Datacomp: In search of the next generation of multimodal datasets." Advances in Neural Information Processing Systems 36 (2024).

[3] Li, Yige, et al. "Anti-backdoor learning: Training clean models on poisoned data." Advances in Neural Information Processing Systems 34 (2021): 14900-14912.

[4] Cheng, Siyuan, et al. "Beagle: Forensics of deep learning backdoor attack for better defense." arXiv preprint arXiv:2301.06241 (2023).

---

### Note · Authors · 2024-11-15

I have read and agree with the venue's withdrawal policy on behalf of myself and my co-authors.